# Influence of the Corporate Image of Nursing Homes on the Loyalty of Residents’ Family Members

**DOI:** 10.3390/ijerph19159216

**Published:** 2022-07-28

**Authors:** Daniel Nadales Rodríguez, Guillermo Bermúdez-González, Ismael Pablo Soler-García

**Affiliations:** Faculty of Commerce and Management, University of Málaga, 29016 Málaga, Spain

**Keywords:** corporate image, consumer behavior, nursing home, loyalty, family decision-making

## Abstract

This study analyzes the influence of the corporate image of nursing homes on the decisions made by family members as to whether their elderly relatives will stay in the same nursing home. An empirical study was conducted considering 566 residents’ family members with the capacity to decide whether said residents will remain in the same nursing home, using a binary regression model with a logistic link function (i.e., logit). For the first time in the nursing home sector, these results show the specific variables of the corporate image that influence family members when deciding whether their elders will stay in the same nursing home. In order of importance, these variables are the level of trust conveyed by the nursing home, the investment made in the facilities, price-quality ratio, emotional connection to the nursing home, and the promotion of the nursing home’s services. The study also highlights the importance of other personal factors in family members’ decisions to keep their elders in the same nursing home, such as the family members’ employment situations (higher loyalty among those employed by third parties) and the determining factors involved in the relative’s choice of nursing home (higher loyalty among those whose choice was mainly based on humane and dignified treatment). This study offers a discussion of the theoretical contributions this research brings to academia as well as managerial implications for the industry. We believe that one future line of research should be continued after the COVID-19 pandemic comes to an end to compare the results and observe whether the most influential variables on family members’ loyalty remain the same as data for this study was collected from November 2019 to February 2020.

## 1. Introduction

According to [1], the world’s population is aging, with the 65+ age group growing the fastest [1]. According to this report, the number of people aged 80 and over will triple from 143 million in 2019 to 426 million in 2050, and one in six people in the world will be over 65, compared to one in eleven in 2019. These percentages are even higher for Europe and North America, where it has been estimated that approximately 25% will be 65 or older. In Spain, these percentages are even more accentuated: it has been estimated that by 2025, 31.4% of the Spanish population will be 65 or older [2].

If the elderly do not have the option preferred by the majority to continue to live at home, one of the most prominent alternatives is moving to a nursing home [3]. Although historically, there have been negative perceptions of nursing homes [4,5,6,7] as impersonal [8] and bureaucratic [9], there is a growing trend for these facilities to not only provide essential care but also to promote Person-Centered Care [8,9,10,11]. This concept entails a series of fundamental principles, including promoting the autonomy and independence of elders, recognizing each person’s uniqueness and interdependence with their environment and family [10,11].

Different studies have shown that close family members usually take the lead in finding a nursing home due to the frailty of their elders [12,13] and that said family members are involved in making decisions regarding their medical care [14,15,16,17]. There are many studies on family members’ emotions and feelings in dealing with this decision-making process, such as betrayal, stress, guilt, sadness, frustration, relief, feeling of not having done enough, etc. [7,15,18,19,20]. However, there are very few studies on family members’ participation in the decision-making process [3,15,16,21,22]. To date, no specific studies have been found on the variables that affect family members’ loyalty to nursing homes and, specifically, family members’ decisions regarding whether their elders will remain in the same nursing home, which is the focus of our research.

The corporate image of nursing homes has also not received sufficient attention in the scientific literature. Even in other sectors, authors that have studied its conceptualization in depth do not agree on the dimensions. It can be defined as the set of impressions, beliefs, and feelings that people have about an organization [23]. Although the positive influence of corporate image on brand, product, or service loyalty has been analyzed in different sectors [24,25,26], its impact has not been empirically tested in the specific sector of nursing homes. It has also not been considered from the perspective of the family members that decide whether their elders will remain in the same nursing home as a focus of the research. This article tries to fill this gap in the literature by analyzing the influence of nursing homes’ corporate image on the decision made by family members (with the authority to make said decision) regarding whether their elders will remain in the same nursing home. The study also analyzes the impact of other variables (such as the reason for choosing the nursing home, the type of nursing home, gender, age, marital status, educational level, and line of work) on the decision of whether elders will remain in the same nursing home.

So, the most recent literature found is very limited: Some of them focus on the residents’ perspective about loyalty or corporate image [27], others are related to factors to influence housing decisions among older adults but not loyalty or corporate image [28,29,30] and others analyze the caregivers or older adults’ experience, strategies or timing of the decision- making process but not loyalty or corporate image [3,4,7,15,16,31,32].

### 1.1. The Corporate Image in Nursing Homes

As the literature has shown, corporate image is a multidimensional concept, according to [23,33], among others, as well as a strategic concept in organizations [34,35]. It is the result of a process by which customers compare and contrast different attributes of companies [36] and is established and developed in the minds of decision-makers through communication and experience [37]. For others, it is the product of all of the experiences, impressions, beliefs, feelings, and knowledge that people have about a company [38] or the result of feelings and beliefs about it [39].

As several authors have pointed out [37,40], as it is becoming increasingly difficult for service companies to differentiate themselves, they try to position themselves by building a strong corporate image that influences customers to choose the company when it is difficult to evaluate the attributes of the services. In addition, for customers with little experience in consuming the services of a certain sector, the image becomes the main driver of future repurchase intention, which is the main indicator of customer loyalty.

There is no unanimous agreement on the most relevant dimensions of corporate image [40,41], although the scientific literature agrees that it should include both functional and emotional aspects [14,41,42,43,44,45] and, specifically, commercial, strategic, social, and emotional dimensions, according to [33,34,46,47], among others. In the specific sector of nursing homes, ref. [29] consider that the more tangible factors related to medical and health care capacity are taken into account more than the social and emotional aspects.

The commercial dimension of the corporate image refers to how the company’s overall marketing is perceived and how customers evaluate the company’s value exchange as compared to others [9,23,48]. This value is therefore determined by factors such as good services [49], the breadth and variety of supply [5,19,21,40,41,50], quality of service [35,49], prices [12,35,49], and accessibility [4,13,51]. It is also important to consider residents’ experience with staff or the quality of interaction with them in this dimension of corporate image [34,35,40,41,43,52,53], including aspects such as the staff’s level of concern [14]; qualifications, professionalism, and effectiveness in solving problems [4,5,31]; warmth and friendliness [14,18,31], and trustworthiness [4,30,35].

On the other hand, the strategic dimension of a corporate image refers to how the overall corporate strategy is perceived, or strategic credibility, as it is referred to by [54]. According to these authors, it is determined by four components: strategic capacity, past corporate performance, how effectively the strategy is communicated, and the credibility or image of senior management. Other authors include innovative character, future projection, and presence in the media in this dimension [46,54]. Likewise, some authors, such as [4], mention equipment, facilities, and their qualified use by staff. Regarding the specific scientific literature on nursing homes, some studies reflect the importance that residents (and their spouses) give to receiving adequate information when selecting a nursing home, which is not always available [4,5,6,13,16,20,32].

The social dimension of the corporate image involves the perception of a company’s social responsibility, which is understood as the company’s concern and commitment to society and the environment through different types of responsible practices [35,41,46,55,56,57], such as sponsoring cultural activities, community involvement and support for social causes [34,37]. As indicated by [41], companies nowadays are not just governed by their economic obligations but also by the goal of reducing the impact of said obligations on their environment. According to [57,58], corporate social responsibility is an important precursor of corporate image and can be a competitive advantage for organizations [57]. In the nursing home sector, ref. [14] highlights the importance family members give to social life in nursing homes for stimulating residents, encouraging interpersonal relationships in the facilities, and staff relationships with residents [18].

Lastly, the emotional dimension of the corporate image affects companies’ positioning in terms of subjective attributes [47]. The corporate image consists of intangible emotional associations and developing feelings and attitudes towards the organization [29,43,59]. According to [41,60], companies also differentiate themselves, within the intangible attributes of the image, by the values and emotions they try to communicate to their target audience or by the degree of sympathy towards the image [37].

### 1.2. Loyalty to Nursing Homes

According to [27], the concept of loyalty is recognized as one of the main elements that helps achieve a company’s goals. As different authors point out, the concept of loyalty can be studied from two aspects: behavioral loyalty and attitudinal loyalty [27,61,62,63]. Behavioral loyalty refers to the frequency of purchases or repurchases made by customers. However, repurchases do not necessarily indicate real loyalty as they may be solely due to the absence of alternatives [37] or out of convenience or habit. Attitudinal loyalty, therefore, emerges as the main indicator of loyalty because it refers to a customer’s psychological and emotional commitment to an organization’s products and services, thereby generating resistance towards opposing opinions, a willingness to pay a premium price, recommending the service to others, and, consequently, their permanence as a customer [26]. According to [62], this attitudinal loyalty creates real loyalty or at least active loyalty that results in not only repurchases but also positive word of mouth. 

The studies conducted by [63] on the nursing home sector and other research in the health care sector [27] underscore the importance of customers’ attitudinal loyalty through adequate communication of the quality of the services provided, thus maintaining long-term competitiveness in the sector. Along the same lines, ref. [25] believe that customer loyalty helps companies reduce costs while increasing sales, trustworthiness, and their competitive advantage over other competitors, with the additional benefit of turning their customers into recommenders [27]. 

### 1.3. The Influence of Corporate Image on the Loyalty of the Elder’s Family Members

As previously mentioned, there are no previous studies considering the loyalty of nursing home residents’ family members. Likewise, no scientific publications were found on the influence of the corporate image of these institutions on the decision of whether to place their elders there. Nevertheless, the positive influence of corporate image on loyalty, both directly and indirectly through other variables, has been widely documented in the service sector. Ref. [37] concluded that corporate image for services that are difficult to evaluate and are only occasionally purchased is very influential on customer loyalty (as is the case in choosing a nursing home). Others, such as [43], showed the relationship between image and loyalty, both directly and indirectly, through variables such as satisfaction with the service provided. Ref. [62] showed that image, in both its functional and emotional dimensions, has a positive influence on customer loyalty. The mediating role of corporate image influencing customer loyalty has also been corroborated in other studies, such as [25], where image acts as a mediator between corporate social responsibility and loyalty, in addition to directly influencing it. Along these lines, refs. [58,64] believe that a good corporate image drives customers to forge stronger ties with companies and recommend them to others, especially when services are difficult to evaluate. 

Ref. [27] have analyzed the precursors of loyalty in the health care sector (similar to the nursing home sector) such as corporate image, in both the attitudinal and behavioral aspects, and have confirmed its direct influence on patient loyalty. 

## 2. Materials and Methods

### 2.1. Description of the Study Location: Spain

With a total of 9.1 million people over the age of 65 at the beginning of January 2019, Spain has the fourth largest elderly population in the European Union, while Germany leads the ranks with 17.9 million. However, speaking in relative terms, Italy tops the ranks at 22.8%, while Spain, at 19.3%, comes in slightly below the EU-27 average of 20.3%, according to [65]. It is important to note that the proportion of elderly people in the EU is increasing yearly. According to population projections, it is estimated that Spain will have more than 14 million elderly people (29.4% of the population) by 2068.

Data for this study were collected from November 2019 to February 2020 using the census of nursing homes in Spain published by [66], which included a total of 5417 nursing homes in April 2019. Of these, 566 were selected by stratified sampling, proportional to the number of nursing homes in each province. The sampling unit consisted of family members of elderly residents who can decide whether their elders will remain in their current nursing homes. The selection of the participants was random, selecting one family member for each nursing home from a list of suggested names provided by the nursing homes.

The questionnaire was completed by 566 participants, one for every nursing home selected. Data were filtered and only 2 outlier or incongruent responses were found. So the final sample size was 564 valid questionnaires.

The participants and nursing home executives did not receive an incentive for participating in the study.

One limitation of this study was the number of responses which, although it was sufficient to give the analysis overall statistical significance for Spain (being the selected region’s representative of the country), it was not sufficient to extract specific aspects in each region.

### 2.2. Sample, Data Collection, and Construct Measures

Data was collected using a 29-item questionnaire. Twenty items were used to measure the family members’ perception of the corporate image (see Table 1), one measured how loyal they were to keeping their elders in the same nursing home (binary variable), while the rest were segmentation variables such as age, gender, marital status, studies, type of work, type of nursing home, and relationship to the elder. The variables related to the commercial dimension of the corporate image were taken from studies by [4,14,23,33,42]. The variables related to the strategic dimension were based on research by [4,33,34,42]. The variables associated with the social dimension were based on research by [46], and the variables associated with the emotional image were taken from studies by [47,60]. All of the variables of nursing homes’ corporate image used a 5-point Likert scale, ranging from Strongly Disagree to Strongly Agree. Loyalty to the nursing home was analyzed through a single dichotomous variable that estimated whether the relative would move the elder to another nursing home, maintaining the same price, as reported in an earlier study in India’s health care sector by [27], where the perception of a fair price is an influential precursor to loyalty, as confirmed by [45,46].

Two questionnaire pretests were conducted as suggested by [67], first through personal interviews and then followed by a small-scale test. A pilot test was also carried out with 12 nursing home executives to identify errors or difficulties in the interpretation of the questions.

### 2.3. Data Analysis

Since the dichotomous dependent variable, that is, the binary variable (yes/no questions), is usually coded as 1 and 0, there are a series of disadvantages of using a linear regression model, including the more than likely non-fulfillment of linear effects [68], the possibility of obtaining estimates below 0 and above 1 [69] and the necessary attribution of special properties for the disturbance parameter [70]. Binary-choice models are normally used to solve these problems [71]. This category includes the well-known logit and probit models, which are available in almost all statistical programs [70]. These models are used in many applied econometric situations [72], in which both are estimated by maximum likelihood [70].

Both models use a latent and, therefore, unobservable variable $y_{i}^{*}$, which represents the net benefit to each individual *i* of a particular course of action as linearly linked to a set of factors *X* and a disturbance process identified by *u* [71]. Keeping in line with [71], the rule for the decision that individuals in the sample did or did not make and the decision could be rewritten as:$$y_{i}=0\;if\;y_{i}^{*}<0$$
$$y_{i}=1\;if\;y_{i}^{*}\geq 0$$

This calculates the likelihood that an individual will take a given course of action in response to a set of stimuli [70], assuming that this action is binary, exhaustive, and mutually exclusive across the entire set of observed individuals [73]. Therefore, applying Baum’s formula [71], the following can be established:$$\mathrm{Pr}\left(y^{*}>0|x\right)=$$
Pr(u>−xβ|x)=
$$\mathrm{Pr}\left(u<x\beta |x\right)=\psi \left(y_{i}^{*}\right)$$
where ψ(·) is a cumulative distribution function (CDF), if the shape of this function is governed by a normal distribution, then the model in question is a probit type, and if the distribution function is logistic, the resulting model will be a logit model [74].

Both cumulative distribution functions usually provide similar results [75,76], especially in the case of univariate binary response models [77], where differences between them are only appreciable in cases of very large samples with extreme patterns in the observed data [78]. The logit model was chosen for this study as it is suitable if a single dependent variable is dichotomous [28], and it is the most popular linking function for modeling binary data [79].

## 3. Results

Table 2 shows the main descriptive statistics of the quantitative variables collected in the questionnaire. With a grand mean of 3.44 and an average deviation of 1.22, the results highlight a certain degree of homogeneity in the average score of the statements presented. In other words, none of the values for the statements have extreme average values (such as very favorable or unfavorable). The item that shows the greatest consensus among respondents on average was IC7, with an average score of 4.048, and IC11 had the highest degree of disagreement. In the case of IC11, it is also important to highlight that its standard deviation of 1.469 is the second highest, exceeded only by IC14 at 1.487, which in turn showed the second highest average disagreement, with a value of 2.723

Table 3 shows the sample profile.

The results show that the sample is aligned with the population profile of the nursing home decision-makers, meaning it is representative. They are mainly children or other family members, women with primary or secondary education, and mostly married, thereby constituting a family group independent from the nursing home user.

Table 4 below breaks down the results of the determining factors of respondents’ nursing home choices.

The most relevant results were medical care and humane and dignified treatment of the elder as the main determining factors in choosing a nursing home. In general, price and proximity to home ranked lower. In addition, the proportion of respondents who said they would stay in the same nursing home was 69.12%, which indicates the overall degree of loyalty.

Due to their conditional nature or the fact that they are common observable variables of reflective constructs, this study performed the procedure ‘backwards’ by sequentially eliminating the least representative variables until the model stabilized. The final model of this process is shown in Table 5.

The overall model is statistically significant (*χ*^2^ = 167.25; *p*-value = 0.000). However, the familiar coefficient of determination or R2 [40] cannot be used to measure the model’s goodness of fit. There are many goodness-of-fit indices to assess the predictive capacity of logit models [80]. In short, the model does not provide enough information to estimate the error term of the latent variable, which is why we have to assume that it is known and the estimation problem does not alter its value in any way [81].

Based on this, several ‘pseudo R2′ goodness-of-fit indices were developed in an attempt to fill that role [80]. McFadden’s Pseudo-R2 [82] is probably the most widely used. The index is also known as the deviance R2 and is likely the simplest one as it attempts to account for both the criterion that is minimized in the estimation as well as the variance calculated by the logistic regression model [80]. As ref. [83] stated, values between 0.2 and 0.4 indicate an excellent fit, which is still a generally accepted benchmark range [84,85]. Accordingly, our model shows the goodness of fit.

Table 6 shows the model’s classification statistics, highlighting its predictive capacity. To summarize, the model can accurately classify approximately 80% of the cases.

After evaluating the overall model, we must assess the impact of the individual predictors. Unfortunately, although the estimated coefficients can be used to test hypotheses, they lack the intuitive interpretation of linear regression estimators [81]. The odds ratios (i.e., the exponential of the regression coefficient) could potentially serve this purpose better and could be evaluated as the effect size [80].

According to the results shown in Table 5, all of the statements regarding the nursing homes’ corporate image that were ultimately kept in the model (i.e., IC4, IC9, IC11, IC13, and IC16) are significant and all of them have a positive influence on loyalty to the nursing home:

-IC9, which refers to the trust generated by the nursing home, is the corporate image variable that has the greatest influence on the loyalty of family members in keeping their elders in the same nursing home. Specifically, for each point (1%), the trust of the decision-makers increases and the probability that the elder will stay in the same nursing home increases by 68.07%.

-IC13, which refers to nursing homes’ investment in their facilities, is the second most influential corporate image variable on the loyalty of family members in keeping their elders in the same nursing home. Specifically, for each point that the decision-maker’s score in this section increases, the probability that family members will decide to change nursing homes decreases by 20.8%. 

-IC4, which refers to the price-quality ratio, is the third most influential corporate image variable on the loyalty of family members in keeping their elders in the same nursing home. Specifically, a one-point change in the price-quality ratio score corresponds to a 19.90% increase in the relative’s loyalty. However, in the robust model, IC4 is no longer significant.

-IC16, which refers to the respondent’s emotional connection to the nursing home, is the fourth most influential corporate image variable on the loyalty of family members in keeping their elders in the same nursing home. Specifically, a one-point increase in the relative’s score in this section corresponds to an 18.06% increase in their loyalty.

-IC11, which refers to the promotion of the nursing homes’ services, is the fifth most influential corporate image variable on the loyalty of family members in keeping their elders in the same nursing home. Specifically, if the relatives value the promotion of the nursing homes’ services by one additional point, the probability that their elders will stay in the same nursing home increases by 17.23%.

Of the remaining variables, the most important predictor that determines changing nursing homes is the respondent’s employment situation. In other words, respondents who are self-employed or employed by third parties are 6 times more likely to change nursing homes than those who are students or homemakers, controlling for all other factors in the model. Likewise, the probability that the elder will remain in the same nursing home is 2.39 times higher for those who state that the primary factor in their decision is humane and dignified treatment than for those who select price.

A similar value was found for those who indicated the elder’s medical care as the primary criterion for their decision. 

It is also important to note that there were no cases in which the distance from home appeared to influence the possibility of staying at or leaving the nursing home. The respondents’ gender, age, marital status, and relationship to the elder also did not affect their decision. The type of nursing home, however, had a major effect: the probability of the elder leaving a public nursing home was 2.79 times higher than for residents in private nursing homes, controlling for all of the other factors in the model.

One of the suggestions for future research related to this article would be to investigate the reasons why the social dimension variables do not influence loyalty as compared to studies in other countries where the results indicate that these variables are actually determining factors. It would also be helpful to conduct a comparative analysis of the perception of residents and family members of the dimensions of the corporate image of nursing homes. 

## 4. Discussion

### 4.1. Theoretical Implications

This study shows interesting findings that, from a theoretical perspective, help expand upon the current knowledge about the corporate image of nursing homes within the context of family members’ decision-making processes regarding whether to keep their elderly relatives in the same nursing home. Specifically, for the first time in the nursing home sector, the corporate image variables that are determining factors in this decision were analyzed. We feel that it is particularly relevant in this specific literature to focus on family members as decision-makers, as this group has been studied very little in the analysis of the relationship between corporate image and loyalty to nursing homes.

Although some studies in the field of nursing homes have analyzed the influence of certain specific variables on relative decision makers, no comprehensive research had previously been conducted to analyze all of the dimensions of the corporate image (commercial, strategic, social, and emotional). This study confirms earlier studies in regard to the importance of some of the specific corporate image variables on residents and their family members.

Specifically, the results of this article are in line with earlier work on the importance of the level of trust conveyed by the nursing home [4,35,42]. The same is true for the importance of the nursing home facilities [40,53], the significance of the appropriate promotion of the residence’s services [5,32,64], the importance of emotional connection to the nursing home [29,59], and the importance of an adequate price-quality ratio [50].

However, unlike other studies, such as those by [34,35,42,46], the social dimension variables of the corporate image were not decisive for family decision-makers when determining whether their elderly relatives would stay in the same nursing home. Variables such as commitment to the environment and society are not among the most important determining factors of loyalty. This may be because in Spain, the nursing homes’ corporate social responsibility actions are valued less or because there is lower social effort in nursing homes in Spain than in other countries. Likewise, our study contradicts other studies, such as those conducted by [14,18], as the social life developed by relatives in the nursing homes was not considered to be a decisive factor for them. We believe it would be of interest to explore these issues in future research further.

Another theoretical contribution of this study is the relationship between the type of nursing home and loyalty, with public nursing homes generating less loyalty in family decision-makers than private nursing homes. We consider this finding to be equally relevant since there is an academic void regarding the impact of the type of nursing home on the loyalty of family members.

### 4.2. Managerial Implications

Based on the results of this study, a series of recommendations can be made for management professionals in the nursing home industry, who are also responsible for their corporate image. This means that if nursing home management professionals want residents to remain at their facilities, they need to take not only the residents into account but also their decision-making relatives in the variables that make up the organization’s corporate image. 

Specifically, within the commercial dimension of corporate image, managers should know how to convey trust (the most influential variable in decision-maker’s loyalty). It is important to pay particular attention to the caregivers of residents that generate trust through their professional skills, personal traits, and communication skills. They should also be aware of the importance the relative decision-maker gives to the balance between the price of the nursing home and the quality of care received.

In regard to the strategic dimension, the corporate image of nursing homes in Spain needs to improve significantly in the two variables that have the greatest influence on the loyalty of family members. In particular, investment in facilities and equipment should be increased while adequately promoting these investments and the associated services addressed in the study by [16].

As for the emotional dimension, management professionals must convey the subjective component of the image of nursing homes by sharing the organization’s values, consequently achieving an emotional connection not only for the residents but also for their family members. 

In our opinion, the fact that the social dimension was not relevant in the family members’ decision to keep their elders in the same nursing home does not mean that nursing homes should abandon their corporate social responsibility actions towards society and the environment. On the contrary, we feel that they should actually make a stronger commitment to society, the environment and to making residents’ social lives more dynamic since one possible reason that this was not a decisive factor may be precisely due to the fact that these actions have not been sufficiently developed in the nursing home sector.

## 5. Conclusions

Although the influence of corporate image on loyalty has been studied extensively, both directly and indirectly, as well as the determining role of customer loyalty in organizations, this study makes an important contribution by specifying which specific corporate image variables determine whether family decision-makers decide to keep their elderly relatives in the same nursing home. In order of importance, these variables are the level of trust conveyed by the nursing home, the investment in its facilities, the promotion of nursing home services, and the emotional connection with the nursing home. Price-quality ratio was also important but lost significance in the robust model.

Therefore, we can conclude that nursing homes need to consider these aspects in their business strategies to positively impact their loyalty, including variables of the commercial, strategic and emotional dimensions of corporate image, while still considering the social dimension. Nursing homes should pay more attention to aspects such as corporate social responsibility and commitment to the environment and society where it provides their services through actions that improve the perception of this dimension of corporate image.

Regarding the other variables, we can also conclude that family members who are self-employed or employed by others are more likely to be less loyal to the nursing homes (as compared to other groups such as retired or unemployed), so nursing homes should pay special attention to this population group in their loyalty strategies.

The study also showed that public sector nursing homes have to make a major effort to improve family decision-maker’s perceptions of their corporate image to retain and increase the loyalty of their residents compared to private nursing homes.

## Figures and Tables

**Table 1 ijerph-19-09216-t001:** Variables of nursing homes’ corporate image.

Variable	Variable Name	Image Dimension
IC1	Services in addition to accommodations	Commercial
IC2	Easy access to the nursing home	
IC3	Quality information and contact with the family	
IC4	Good price-quality ratio	
IC5	Correct advice from management before making decisions	
IC6	Quick problem solving	
IC7	Kindness in staff treatment of residents	
IC8	Effort to provide quality care	
IC9	Transmission of trust	
IC10	Professional qualifications	Strategic
IC11	Promotion of the nursing home services	
IC12	Innovation in treatments and services	
IC13	Investment in facilities and equipment	
IC14	Growth projection	
IC15	Proper management of the nursing home	
IC16	Emotional connection to the nursing home	Emotional
IC17	Transmission of enthusiasm	
IC18	Commitment to improving society	Social
IC19	Social life in the nursing home	
IC20	Commitment to the environment	

Source: Prepared by the authors.

**Table 2 ijerph-19-09216-t002:** Descriptive statistics.

Variable	Obs	Mean	Std. Dev.
IC1	566	3682	1072
IC2	566	3171	1293
IC3	565	3950	1001
IC4	564	3559	1127
IC5	564	3601	1235
IC6	564	3605	1075
IC7	564	4048	1041
IC8	564	3927	1096
IC9	564	3956	1054
IC10	564	3839	1117
IC11	564	2679	1469
IC12	564	2970	1326
IC13	564	3074	1353
IC14	564	2723	1487
IC15	564	3353	1360
IC16	564	2924	1398
IC17	564	3087	1309
IC18	564	3129	1352
IC19	564	4234	0942
IC20	564	3092	1414

Source: Prepared by the authors.

**Table 3 ijerph-19-09216-t003:** Sample composition.

Variable	Modality	Freq	Percentage
Age	Under 35	73	12.97
	35–45	83	14.74
	46–55	178	31.62
	56–65	120	21.31
	Over 65	109	19.36
Relationship	Child	220	38.87
	Partner	40	7.07
	Sibling	84	14.84
	Other family member	222	39.22
Marital Status	Single	78	14.13
	Married	326	59.06
	Separated/Divorced	75	13.59
	Widower	73	13.22
Educational Level	Primary School	187	33.75
	Secondary School	212	38.27
	University	155	27.98
Type of work	Employee	196	35.06
	Self-employed	75	13.42
	Unemployed	52	9.3
	Retired	107	19.14
	Student	32	5.72
	Homemaker	97	17.35
Gender	Man	234	41.94
	Woman	324	58.06

Source: Prepared by the authors.

**Table 4 ijerph-19-09216-t004:** Determining factors in the nursing home choice.

Variable	Modality	Freq	Percentage
Type of nursing home in which the elder is staying	Public	245	43.36
	Private	299	52.92
	Charitable	21	3.72
PRIMARY determining factor in nursing home choice	The price of the nursing home	84	15.03
	The elder’s medical care	178	31.84
	Humane and dignified treatment of the elder	236	42.22
	Proximity of the elder to my home	61	10.91

Source: Prepared by the authors.

**Table 5 ijerph-19-09216-t005:** Logit regression results.

Would Change	Coef	Odds Ratio	Std. Err.	*p* > |z|		Robust Std. Err.	Robust *p* > z	
IC 4	−0.181	0.834	0.110	0.099	*	0.111	0.101	
IC 9	−0.519	0.595	0.125	0.000	***	0.130	0.000	***
IC 11	−0.159	0.853	0.087	0.066	*	0.093	0.087	*
IC13	−0.234	0.792	0.094	0.013	**	0.098	0.017	**
IC 16	−0.166	0.847	0.097	0.086	*	0.099	0.095	*
**Relationship**								
Partner	0.013	1.013	0.545	0.981		0.527	0.98	
Sibling	0.191	1.210	0.411	0.643		0.422	0.651	
Other family member	−0.172	0.842	0.324	0.595		0.320	0.591	
**Type of residence**								
Private	−1.024	0.359	0.243	0.000	***	0.243	0.000	***
**Charitable**	0.047	1.048	0.580	0.936		0.542	0.932	
Chose								
No	0.228	1.256	0.274	0.405		0.265	0.389	
**DE1**								
The elder’s medical care	−0.723	0.485	0.376	0.055	*	0.344	0.035	**
Humane and dignified treatment of the elder	−0.871	0.419	0.368	0.018	**	0.374	0.02	**
Proximity of the elder to my home	−0.663	0.515	0.465	0.153		0.426	0.12	
**Gender**								
Woman	−0.361	0.697	0.261	0.167		0.270	0.182	
**Age**								
From 35 to 45	−0.179	0.836	0.568	0.753		0.509	0.726	
From 46 to 55	−0.411	0.663	0.576	0.475		0.482	0.393	
From 56 to 65	−0.585	0.557	0.620	0.346		0.534	0.273	
Over 65	−0.749	0.473	0.750	0.318		0.707	0.29	
**Marital Status**								
Married	0.184	1.202	0.495	0.71		0.448	0.682	
Separated/Divorced	0.827	2.285	0.567	0.145		0.527	0.117	
Widower	0.501	1.650	0.632	0.428		0.599	0.403	
**Educational level**								
Secondary School	−0.581	0.559	0.317	0.067	*	0.318	0.068	*
University	−0.532	0.588	0.378	0.16		0.393	0.176	
**Work**								
Employed by third party	1.800	6.049	0.798	0.024	**	0.998	0.071	*
Self-employed	1.826	6.206	0.873	0.037	**	1.056	0.084	*
Unemployed	1.679	5.362	0.874	0.055	*	1.048	0.109	
Retired	1.743	5.717	0.933	0.062	*	1.114	0.118	
Homemaker	1.356	3.880	0.878	0.122		1.071	0.206	
CONS	4.368	78.890	1.002	0.000	***	1.063	0.000	***

Number of observations: 523; Likelihood-Ratio (Chi2 [33df]): 167.25; Prob > Chi2: 0.000; Pseudo R2: 0.2577; Significant at 1% (***), 5% (**), or 10% (*). The results reported are from the ‘logit’ command (Stata). Source: Prepared by the authors.

**Table 6 ijerph-19-09216-t006:** Classification values.

Classified	True		
	D	~D	Total
+	89	31	120
−	74	329	403
Total	163	360	523
Classified + if predicted Pr(D) ≥ 0.5			
True D defined as WOULD CHANGE! = 0			
Sensitivity		Pr(+D)	54.60%
Specificity		Pr(−~D)	91.39%
Positive predictive value		Pr(D+)	74.17%
Negative predictive value		Pr(~D−)	81.64%
False + rate for true ~D		Pr(+~D)	8.61%
False-rate for true D		Pr(−D)	45.40%
False + rate for classified +		Pr(~D+)	25.83%
False-rate for classified−		Pr(D−)	18.36%
Correctly classified			79.92%

Source: Prepared by the authors.

## Data Availability

Not applicable.

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
