# Peer review of "Influence of the Corporate Image of Nursing Homes on the Loyalty of Residents’ Family Members"

_ijerph, 2022, doi:10.3390/ijerph19159216_

Round 1

Reviewer 1 Report

Influence of the Corporate Image of Nursing Homes on the 2 Loyalty of Residents’ Family Members

The study raises an interesting issue and is a good step towards learning more about the impact of the image of nursing homes on the placement of relatives.

The theoretical background is broadly presented and extensive literature is cited. Conclusions follow logically from the results. The discussion could refer to other studies more broadly.

Research limitations are presented.

The text needs to be edited in line with the editorial requirements.

Author Response

Thank you very much for your instructions.

We have proceeded to correct, in line with the editorial requirements: 

1- the sections.

2-the references have been numbered in order of appearance in the text and listed individually at the end of the manuscript. 

We believe that the rest of the requirements are met. Thank you very much and regards.

Reviewer 2 Report

Overall an interesting paper, which examines / analyzed the influence regarding the nursing home corporate image and the decisions made by family members regarding elderly relatives remaining at the nursing home facility using a binary regression model with logistic link function with 566 Spanish respondents. From this perspective, the paper will garner interest from both academia and c-suite industry leaders as well as policy makers. From an academic perspective the mix of current (less than 5 year peer reviewed research) is not fully evident in the paper which impacts the comprehensiveness of the paper, as well, there is a lack of divergent and conflicting perspectives. In my opinion, and experience, scholars expect journal articles which present recent research and an inclusion of divergent and conflicting perspectives as opposed to a narrow focus. As well, c-suite industry and policy makers would have a similar expectation. On a positive note, while somewhat dated, the author(s) have provided a overall germane discussion with the literature – however, as noted previously, I would have personally liked to see an inclusion of additional recent and divergent discussion of conflicting findings and/or theoretical positions causing intellectual tension in the field, which is currently absent - I feel this is a shortcoming and I believe the academic community and others will consider it a necessary inclusion to the paper to ensure the paper’s overall comprehensiveness and currency to the intended audience (I recommend identifying 4 -6 current articles; and two – three paragraphs related to conflicting and divergent perspectives).  

From a methodology perspective, the paper is generally well designed and appropriate – the exception is the lack of limitations at the front end of the paper; and this may very well reflect my own personal preferences. However, in defense of this posture, the identification of limitations at the front (Abstract and within the Methodology Section) enables the reader audience to understand the limitations and any caveats up front, and prior to reading the paper through and then findings limitations at the back end of the paper. It is imperative that reader(s) have the opportunity to place the paper into a context in relation to what is stated / proposed by the author(s). As well, clarify how many questionnaires were distributed, the number of returned, the number of unusable questionnaires. Did participants receive an incentive or did nursing home executives receive an incentive. How were participants selected – from a full and complete list provided by each nursing home or from a list of suggested names provided by the nursing home. Are the selected regions representative of Spain – if not to what degree is the study representative of the aged population.

The results are presented in a clear and concise manner; with a depth, breath and scope of data analysis evident. the inclusion of Tables is an effective visual. As well, the author(s) have developed an acceptable linkage between the results and conclusions noted. The points noted in paper are tied together into a final coherent picture. It is evident that the author(s) have an excellent understanding of the subject area.

This is a solid paper in many respects, since it provides several opportunities for continued research in the subject area with the possibility of different streams within the research area, while providing further avenues of research potential. With respect to the practical application of the research, it presents an opportunity to enhance the depth, breadth and understanding the loyalty and trust associated with nursing homes; and the potential impact related to perceptions of facilities maintenance, self promotion / marketing of facilities by nursing homes / services.  

The writing quality of the paper is grammatical correct, which in turn demonstrates scholarly flow and readability. A professional English edit is not required.

For consideration, I recommend the inclusion of research which presents / discusses current (less than 5 years old) and a brief inclusion of the conflicting and divergent perspectives in the literature (2 -3 paragraphs), and to further updated the Abstract / Methodology to note the limitations to your paper as opposed to the back end of the paper. As well, there is a need to clarify several issues related to the methodology in the paper. If such steps are taken, you will gain global interest in your paper, and a following for your research. 

Author Response

Thank you very much for your instructions.

We have proceeded to correct the identification of limitations at the front (abstract and within the methodology Section) as opposed to the back end of the paper  (lines 23 to 26 and lines 410 to 415).

The clarification of how many questionnaires were distributed, the number of returned, the number of unusable questionnaires, and so on have been done (lines 198 to 214).

 In relation to include recent research, we haven´t found recent articles about healt care sector , relatives and loyalty/corporate image, but we have included a new paragraph (lines 71 to 76) : “So, the most recent literature found is very limited: Some of them focus on the residents´ perspective about loyalty or corporate image [27], others are related to factors to influence housing decisions among older adults but not loyalty or corporate image [28,29, 30] and others analyze the caregivers or older adults´experience, strategies or timing of the decision- making process but not loyalty or corporate image [3, 4, 7, 15, 16, 31, 32]”  

Six of the eleven references are less than 5 year and the rest are less than 10 years. 

In relation to divergent and conflicting perpectives we included in the previous version (lines 439 to 448): “However, unlike other studies, such as those by [35, 36, 43, 47], the social dimension variables of corporate image were not decisive for family decision-makers when determining whether their elderly relatives would stay in the same nursing home …“ 

Thank you very much and regards